# Oncologic Outcomes of Surgically Treated Cervical Cancer with No Residual Disease on Hysterectomy Specimen: A 4C (Canadian Cervical Cancer Collaborative) Working Group Study

Christa Aubrey [1,*], Gregory R. Pond [2], Limor Helpman [3], Danielle Vicus [4], Laurie Elit [3], Marie Plante [5], Susie Lau [6], Janice S. Kwon [7], Alon D. Altman [8], Karla Willows [9], Tomer Feigenberg [10], Jeanelle Sabourin [1], Vanessa Samouelian [11], Laurence Bernard [3], Norah Cockburn [3], Nora-Beth Saunders [8], Sabrina Piedimonte [4], Ly-Ann Teo-Fortin [5], Soyoun Rachel Kim [4], Noor Sadeq [9], Ji-Hyun Jang [7], Sarah Shamiya [12] and Gregg Nelson [13]

[1] Division of Gynecologic Oncology, Cross Cancer Institute, University of Alberta, Edmonton, AB T6G 1Z2, Canada
[2] Department of Oncology, Escarpment Cancer Research Institute, McMaster University, Hamilton, ON L8V 5C2, Canada
[3] Division of Gynecologic Oncology, Juravinski Cancer Center, McMaster University, Hamilton, ON L8V 5C2, Canada
[4] Division of Gynecologic Oncology, Sunnybrook Health Science Center, University of Toronto, Toronto, ON M4N 3M5, Canada
[5] Division of Gynecologic Oncology, Centre Hospitalier Universite de Quebec, Laval University, Quebec City, QC G1R 2J6, Canada
[6] Division of Gynecologic Oncology, Jewish General Hospital, McGill University, Montreal, QC H3T 1E2, Canada
[7] Division of Gynecologic Oncology, Vancouver General Hospital, University of British Columbia, Vancouver, BC V5Z 1M9, Canada
[8] Division of Gynecologic Oncology, Winnipeg Women's Hospital, University of Manitoba, Winnipeg, MB R3E 0L8, Canada
[9] Division of Gynecologic Oncology, Queen Elizabeth II Health Science Center, Dalhousie University, Halifax, NS B3K 4N1, Canada
[10] Division of Gynecologic Oncology, Trillium Health Partners, Mississauga, University of Toronto, Toronto, ON L5B 1B8, Canada
[11] Division of Gynecologic Oncology, Centre Hospitalier de l'Universite de Montreal, University of Montreal, Montreal, QC H2X 3E4, Canada
[12] Department of Obstetrics and Gynecology, University of Alberta, Edmonton, AB T5H 3V9, Canada
[13] Division of Gynecologic Oncology, Tom Baker Cancer Center, University of Calgary, Calgary, AB T2N 4N2, Canada
* Correspondence: christa.aubrey@albertahealthservices.ca; Tel.: +1-780-432-8337

**Abstract:** Minimally invasive surgery for the treatment of macroscopic cervical cancer leads to worse oncologic outcomes than with open surgery. Preoperative conization may mitigate the risk of surgical approach. Our objective was to describe the oncologic outcomes in cases of cervical cancer initially treated with conization, and subsequently found to have no residual cervical cancer after hysterectomy performed via open and minimally invasive approaches. This was a retrospective cohort study of surgically treated cervical cancer at 11 Canadian institutions from 2007 to 2017. Cases initially treated with cervical conization and subsequent hysterectomy, with no residual disease on hysterectomy specimen were included. They were subdivided according to minimally invasive (laparoscopic/robotic (MIS) or laparoscopically assisted vaginal/vaginal hysterectomy (LVH)), or abdominal (AH). Recurrence free survival (RFS) and overall survival (OS) were estimated using Kaplan–Meier analysis. Chi-square and log-rank tests were used to compare between cohorts. Within the total cohort, 238/1696 (14%) had no residual disease on hysterectomy specimen (122 MIS, 103 AH, and 13 VLH). The majority of cases in the cohort were FIGO 2018 stage IB1 (43.7%) and underwent a radical hysterectomy (81.9%). There was no statistical difference between stage, histology, and radical vs simple hysterectomy between the abdominal and minimally invasive groups. There were no significant differences in RFS (5-year: MIS/LVH 97.7%, AH 95.8%, $p = 0.23$) or OS (5-year: MIS/VLH

98.9%, AH 97.4%, *p* = 0.10), although event-rates were low. There were only two recurrences. In this large study including only patients with no residual cervical cancer on hysterectomy specimen, no significant differences in survival were seen by surgical approach. This may be due to the small number of events or due to no actual difference between the groups. Further studies are warranted.

**Keywords:** cervical cancer; minimally invasive surgery

## 1. Introduction

The optimal surgical treatment for early-stage cervical carcinoma has evolved, and circled back to traditional approaches over the last several years. With the advent of minimally invasive surgical (MIS) techniques, the abdominal radical hysterectomy was challenged. Extrapolating safety from other tumor sites, the MIS approach for radical hysterectomy demonstrated favorable length of stay and recovery outcomes [1]. Subsequently, as robotic surgery gained traction, the minimally invasive radical hysterectomy continued to grow in popularity [2,3]. However, in 2018 the first prospective randomized trial which examined the safety of open versus laparoscopic/robotic radical hysterectomy (Laparoscopic Approach to Cervical Cancer—LACC Trial) was published and found lower rates of disease-free and overall survival with the minimally invasive approach [4]. From this trial stemmed a myriad of discussions around the role of minimally invasive surgery and cervical carcinoma, and what populations (if any) could MIS be a safe approach [5].

In the LACC trial, the majority of patients included were stage IB1 (FIGO 2009 staging), with stage IA1 with LVSI and IA2 patients making up <10% in each of the study arms [4]. Accordingly, the extrapolation of the LACC trial findings for microscopic cervical tumors has been questioned, and the safety of MIS in these cases has been postulated [6]. Furthermore, in the wake of the LACC trial there has been much effort to determine which factors impact the risk of recurrence after surgery, apart from tumor size, or stage. In a recent multi-institutional retrospective study of laparoscopic radical hysterectomy for stage IA1-1B1 tumors, tumor size and extent of residual disease at the time of surgery were main independent predictors of recurrent disease postoperatively, and that preoperative conization reduced the risk of recurrence [7]. Preoperative conization has been suggested as a possible protective maneuver prior to surgery [8]. The question remains whether certain microscopic cases of cervical cancer, with lower likelihood of residual disease at the time of hysterectomy [9], could be considered for a minimally invasive surgical approach. Given the specific population of interest and relatively low event rate, prospective studies focusing on this question are challenging. Hence, large, multi-site, retrospective reviews are necessary to help patients and clinicians make informed decisions on treatment options.

The objective of this study was to compare the oncologic outcomes in cases treated with cervical conization and subsequent hysterectomy, with of no residual cervical cancer found in the hysterectomy specimen performed via open and minimally invasive approaches using the multicenter Canadian Cervical Cancer Collaborative (4C) database.

## 2. Materials and Methods

The Canadian Cervical Cancer Collaborative represents a database of 11 tertiary institutions across Canada, capturing all cases of surgically treated cervical carcinoma from January 2007 to December 2017. All participating sites had approval from their local Research Ethics Boards.

Inclusion criteria were age > 18 years old, squamous, adenocarcinoma, or adenosquamous histology cervical carcinoma, and primary surgical treatment. Surgical approach included both simple and radical hysterectomies performed by laparotomy, laparoscopic or robotic, or laparoscopically assisted vaginal or vaginal hysterectomy. Only patients with nodal evaluation (sentinel lymph node biopsy or lymphadenectomy) were included in the present study. For this study, all patients had to have no residual carcinoma on

final hysterectomy specimen. Indication for excisional procedure was not collected, but all cases had a diagnosis of cervical carcinoma prior to definitive surgical treatment. Cases were excluded if they underwent neoadjuvant radiation therapy or chemotherapy, were stage IA1 without LVSI, stage IV, underwent trachelectomy or conization as their definitive surgery, tumors > 4 cm (by clinical exam or imaging- pre-conization), "other" histologies, or with missing pertinent information (tumor size, surgical type, final stage).

Staging was based on the pathological FIGO 2018 staging system. Recurrences were determined on the basis of either radiographic evidence or biopsy-proven recurrence. Follow up was in accordance with local standard of care. Data were collected locally at each institution and entered into a central REDCap database. Data collected included demographics (age, smoking status, ASA score, height, weight and BMI calculation), preoperative imaging (MRI and/or PET scan), operative factors (mode of surgery, lymph node assessment, uterine manipulator, complication), postoperative factors (length of stay, readmission), pathologic characteristics (FIGO 2018 pathologic staging, histology, tumor diameter, stromal invasion, lymphovascular space invasion, lymph node status, and adjuvant treatment (chemotherapy and radiotherapy), along with recurrence and follow up data.

Descriptive statistics were used to summarize demographic, treatment, and disease variables, as well as outcomes. Chi-square, Wilcoxon rank sum, and log-rank tests were used to explore for differences in characteristics based on surgical technique, as well as the potential effect of surgery on outcomes of interest. Recurrence free survival (RFS) and overall survival (OS) were estimated using the Kaplan–Meier method. Given the small number of disease-specific events, no regression analysis was performed. All analyses and confidence intervals were two-sided and statistical significance was defined at the $\alpha = 0.05$ level. SAS version 9 statistical software package was used for analysis.

## 3. Results

The entire 4C database of primary surgically treated cervical carcinomas was comprised of 1696 cases. After excluding cases that were stage IA1 and negative LVSI, stage IV, unknown tumor stage, tumors > 4 cm, and those with unknown tumor size and stages IIA, IIB, or IIIC there were 1342 cases remaining. Radical or simple trachelectomy/conization for definitive surgery, and cases with no lymph node sampling or missing surgical type were then excluded, leaving 1170 cases. Finally, those with missing disease status, residual disease at surgery, or other histology were then excluded. This left 238 cases with no residual disease on hysterectomy specimen included in this analysis, and represented 14.0% of the entire 4C cohort.

Descriptive demographic, operative, pathologic, and treatment variables are summarized in Table 1. Median age was 42, and the most common histology was squamous (58.8%), with adenocarcinomas representing 38.2% of the cohort. The final stage was predominantly 1B1 (43.7%), with 5 patients having positive lymph nodes (2.1%). Over a third did not undergo any preoperative imaging (39.9%). Minimally invasive surgical approaches included robotic, laparoscopic, and combined vaginal/laparoscopic, accounting for 56.7% of the cohort, with the majority (95.8%) having pelvic node assessment only (either sentinel or full lymphadenectomy). A uterine manipulator was used in a minority of cases (17.7%). The majority of cases undergoing MIS and abdominal hysterectomy were stage IB (40.7% and 47.6%) and underwent a radical hysterectomy (80.7% and 83.5%). The median tumor size was 7.2 mm, with 22.7% having lymphovascular space invasion. Additionally, 89.5% did not receive any adjuvant treatment.

**Table 1.** Baseline demographic variables.

| Variable | Sub-Variable | N |
|---|---|---|
| Total cohort | | 238 |
| Age | Median (IQR) range | 42 (36, 49), 21–84 |
| Histology | Squamous Cell | 140 (58.8) |
| | Adenocarcinoma | 91 (38.2) |
| | Adenosquamous | 7 (2.9) |
| Stage | N (%) IA1 | 55 (23.1) |
| | IA2 | 68 (28.6) |
| | IB1 | 104 (43.7) |
| | IB2 | 6 (2.5) |
| | IIIC1 | 5 (2.1) |
| Smoking Status | N (%) Never Smoker | 94 (39.5) |
| | Ex-Smoker | 28 (11.8) |
| | Current Smoker | 55 (23.1) |
| | Missing | 61 (25.6) |
| BMI | Median (range), n | 26.9 (17.2, 87.0), 163 |
| Type of Pre-Op Imaging | N (%) MRI† | 68 (28.6) |
| | CT | 34 (14.3) |
| | PET/CT | 38 (16.0) |
| | None | 95 (39.9) |
| | Missing/Unknown | 7 (2.9) |
| Tumour Size (mm) | Median (IQR), range | 7.2 (3.5, 12), 0–40 |
| Surgery Type | N (%) Robotic | 44 (18.5) |
| | Laparoscopy | 78 (32.8) |
| | Open | 103 (43.3) |
| | Combined vaginal/laparoscopic | 13 (5.5) |
| Open Conversion | N (%) Yes | 6/131 (4.6) |
| Cervical Surgery Type | Radical hysterectomy | 195 (81.9) |
| | Simple hysterectomy | 43 (18.1) |
| Lymph Node Surgery | Pelvic Node Assessment | 228 (95.8) |
| | Pelvic and Para-aortic Node Assessment | 10 (4.2) |
| Use of Intra-Uterine Manipulator | No | 179 (75.2) |
| | Yes | 42 (17.7) |
| | Missing | 17 (7.1) |
| Length of Stay | Median (range), n | 2 (0, 30), 177 |
| Lymphovascular space invasion | positive | 54 (22.7) |
| | negative | 127 (53.4) |
| | Not reported | 57 (24.0) |
| Post-Op Treatment | None | 214 (89.9) |
| | Radiation | 5 (2.1) |
| | Chemo-radiation | 5 (2.1) |
| | Not reported | 14 (5.9) |

The distribution of variables between surgical approach is shown in Table 2. We found no difference in stage, tumor size, histology, smoking status, or those undergoing simple versus radical hysterectomy. In those with adenocarcinomas, there was a 9.6% difference in open vs minimally invasive approach, but this was not statistically significant. There was a statistically different distribution in age for patients undergoing open versus minimally invasive surgery (median age of 45 for open versus 41 for minimally invasive, $p = 0.004$). Additionally, the BMI of patients undergoing open surgery was higher than the minimally invasive approach (median BMI 28.8 for open, versus 25.6 for minimally invasive, $p = 0.015$).

Length of stay was also significantly different between open and minimally invasive cases (median 4 days in open versus 1 in minimally invasive, *p* < 0.001). The impact of surgical approach on 5-year OS and RFS and was not statistically significant.

**Table 2.** Characteristics of patients grouped by surgical approach.

|  | * MIS/VLH | Open | *p*-Value |
|---|---|---|---|
| N | 135 | 103 | N/A |
| Median (IQR) Age | 41 (36, 46) | 45 (38, 55) | 0.004 |
| Stage IA1 | 32 (23.7) | 23 (22.3) | |
| IA2 | 42 (31.1) | 26 (25.2) | |
| IB1 | 55 (40.7) | 49 (47.6) | 0.83 |
| 1B2 | 3 (2.2) | 3 (2.9) | |
| IIIC1 | 3 (2.2) | 2 (1.9) | |
| Squamous Cell | 86 (63.7) | 54 (52.4) | |
| Adenocarcinoma | 46 (34.1) | 45 (43.7) | 0.20 |
| Adenosquamous | 3 (2.2) | 4 (3.9) | |
| N (%) Never Smoker | 48 (35.6) | 46 (44.5) | |
| Ex-Smoker | 19 (14.1) | 9 (8.7) | |
| Current Smoker | 35 (25.9) | 20 (19.4) | 0.27 |
| Missing | 33 (24.4) | 28 (27.2) | |
| Median (IQR) BMI | 25.6 (22.7, 30.8) | 28.8 (23.3, 34.7) | 0.015 |
| Median (IQR), Tumour Size (mm) | 7.3 (3.5, 11) | 7.5 (3, 12) | 0.89 |
| N (%) radical hysterectomy | 109 (80.7) | 86 (83.5) | 0.58 |
| Median (IQR) Length of Stay | 1 (0, 2) | 4 (3, 5) | <0.001 |
| N (%) Deaths 5-year (95% CI) OS | 1 (0.7) 98.9 (92.5, 99.8) | 5 (4.8) 97.4 (90.1, 99.3) | 0.10 |
| N (%) Events 5-year (95% CI) RFS | 2 (1.5) 97.7 (91.1, 99.4) | 5 (4.8) 95.8 (87.4, 98.6) | 0.23 |

* Surgical Approach: MIS/LVH: Robotic/Laparoscopy or laparoscopically assisted vaginal/vaginal.

In the cohort, there were six deaths, but only one was due to disease. The overall 5-year OS was 98.4% (95% CI 95.0, 99.5) and 5-year RFS was 96.6% (95% CI 92.5, 98.5). There were two recurrences, one who died of disease, and the other alive with disease at time of data collection (Table 3). One recurrence was initially stage IB2, had an open surgical approach, and recurrence was in the lymphatics which resulted in death from disease. The other recurrence was initially stage IB1, had a laparoscopic surgical approach, and recurrence was in the vagina. This case was alive with disease at time of data collection. Neither of these cases received adjuvant treatment after surgery. Since there were only two disease-specific events, regression analysis was not feasible.

**Table 3.** Recurrence details.

| Patient | Age | Stage | Surgical Approach | Histology | Recurrence | Site of Recurrence | Status * | Time to Recurrence (Months) |
|---|---|---|---|---|---|---|---|---|
| 1 | 64 | IB2 | Open | Adeno | Yes | Lymphatic | DOD | 39.9 |
| 2 | 57 | IB1 | Laparoscopic | Squamous | Yes | Vaginal | AWD | 37.8 |

* Status: DOD: Dead of Disease; AWD: Alive with disease.

## 4. Discussion

In this retrospective cohort study from 11 academic institutions in Canada over a 10-year time span, we captured data on surgically treated cervical carcinoma. For this study

we specifically looked at the outcomes in those females who had no residual carcinoma on final hysterectomy specimen: 238 cases, representing 14.0% of the entire 4C cohort. The majority were FIGO 2018 stage 1B1, and a small number (2.1%) had positive pelvic lymph nodes. The majority of patients (89.9%) did not receive adjuvant treatment. There were only two recurrences, and no differences observed based on surgical approach on OS or RFS in this cohort, with a 5-year OS exceeding 95% regardless of surgical technique.

This cohort of patients had no residual carcinoma on hysterectomy specimen. In keeping with data from a retrospective multi-institutional review of early-stage cervical carcinoma in 815 patients, where in those with a prior cone and no residual tumor on preoperative assessment (243 patients), there was no differences between the open and minimally invasive treated rates of recurrences (1.4% vs. 2.9%, not significant) [10]. Similarly, in the recent final analysis of the LACC trial, there was no difference in OS demonstrated between the open versus MIS group in those with no residual tumor on final hysterectomy specimen [11]. Casarin et al., in a multi-institutional study of 428 patients, found that residual tumor on final pathologic specimen was shown to be the strongest predictor of risk of recurrence with an Odds Ratio (OR) of 5.29, with a preoperative cone reducing the risk of recurrence (OR 0.32) [7]. Similarly, Zanagnolo et al. described a single institution study of 198 patients treated with robotic radical hysterectomy where 116 (58.6%) had a prior conization, and in cases where there was no residual disease on hysterectomy specimen, there was a statistically significant improvement in PFS, with no recurrences in this group [12].

There is a growing body of literature around surgical prognostic features associated with recurrence of cervical carcinoma. In the recently published Surgery in Cervical Cancer Comparing Different Surgical Approaches in Stage IB1 Cervical Cancer (SUCCOR) cone study, preoperative conization was demonstrated to be a favorable prognostic factor for recurrence (65% reduction) and survival outcomes (75% reduction in risk of death) after both open, and minimally invasive surgery for cervical carcinoma [8]. Additionally, in the recently presented final LACC analysis, there were no differences in DFS between the MIS and open group in those with a preoperative conization (HR 1.27 (0.39–4.17), $p = 0.69$) [11]. These findings have led to the postulation of preoperative excisional procedures being a protective maneuver, and if even a minimally invasive approach could be safely considered in this group of patients.

However, as discussed in the accompanying editorial, 80.7% of cases in the SUCCOR cone study had residual disease on hysterectomy specimen, much higher than would be expected in those who qualify for a preoperative excisional procedure, and therefore circumstances and indications for conization are unclear [13]. In a retrospective study looking at 198 cases of stage IA to IB cervical carcinoma who had an initial cervical conization followed by hysterectomy, residual carcinoma was found in 78 females (39.4%), and on the initial cone specimen in this study there were only 36 that had negative margins (18.8%) [14]. Negative conization margin has been shown to be a significant predictor of no residual disease at the time of hysterectomy [9,14,15]. In a propensity-matched cohort of 70 patients who had an initial conization with negative margins, and subsequent radical hysterectomy either MIS or open, there were five cases (14.3%) who had residual disease on hysterectomy specimen, and surgical approach in this study did not influence recurrence, with only one recurrence in each group [16]. In the ConCerv trial, negative margins for invasive and pre-invasive disease were inclusion criteria for enrollment in the study, and in this study, only 1/40 (2.5%) of cases that underwent hysterectomy had residual disease at the time [17]. Therefore, in those cases with preoperative conization with negative margins, there is a low likelihood of residual disease on hysterectomy specimen, and represent a low-risk group in which the role of surgical approach on disease outcomes should be better clarified.

Additionally, in our study, the rate of lymph node positivity was 5/238 (2.1%), which is lower than reported in the ConCerv trial with an incidence of 5% [17]; however, our population did include stage IA1 tumors with LVSI, representing 23.1% of the cohort. The risk/benefit ratio of lymph node evaluation must be taken into account. In our study, there

were five cases (2.0%) of major intraoperative bleeding, and only three cases (1.2%) of major postoperative complications. Additionally, only 16.4% of the cohort had sentinel lymph node evaluation, which is reflective of the time-frame of data collection. Therefore, major intraoperative and postoperative morbidity was low, and as the sentinel lymph node approach becomes more adopted, this will likely be even lower. Support for sentinel lymph node evaluation in this low-risk population, rather than full lymphadenectomy, is also supported a large retrospective study of 463 cases of early cervical cancer, in which an algorithm for determining risk of lymph node involvement was identified [18]. In this context, the importance of lymph node sampling remains substantial, as consequences of positive lymph nodes in terms of recommendations for adjuvant treatment are significant, and even in this group of no residual disease on hysterectomy there were still four cases of positive pelvic lymph nodes.

The main strength of this study is it being a large, multi-institutional collective effort to identify consecutive cases of surgically treated cervical carcinoma in Canada from December 2007 to January 2017 [6]. This dataset includes patient level data, and reflects a period of time where a variety of surgical approaches were used in treatment of early-stage cervical disease. Cases were individually identified and raw data were extracted from the chart and pathology reports. Additionally, the data set was queried as a whole, and individual cases requiring further clarification were available. In focusing of cases who first underwent a conization, and then were followed by hysterectomy where no residual disease was identified, we further characterized this low-risk population.

Limitations include the retrospective nature of the study, and the low event rate of recurrence. Conclusions cannot be definitively drawn, but should be considered hypothesis-generating. Particularly, we were interested in cases with no residual carcinoma on hysterectomy specimen, but specifics on the excisional specimens, including indication for conization, margin status for invasive or pre-invasive disease, and number of passes for excisional procedures were not accurately captured in the database. Additionally, there was no centralized pathology review. The time in which the data were collected (2007–2017) reflects a period with advances in imaging, surgical procedures, pathologic reporting, and adjuvant treatment occurred, which can make interpretation of data challenging. Further, in this retrospective dataset surgical approach was determined by the attending physician, and procedural differences could not be accounted for. The goal was to capture all cases within the defined period of time; however, since follow up was based on local practice and standard of care there could have been recurrences that were missed due to this or relocation.

There is interest in identifying specific prognostic features that may impact surgical outcomes in the management of cervical carcinoma. In particular, the impact of preoperative conization as a protective maneuver has been proposed. It is hypothesized that preoperative excisional procedures may translate into more favorable outcomes, both in the minimally invasive and open surgical modalities [7–10,19–21]. However, nuances of this benefit are still not well understood. Many studies have focused on the excisional procedure as the variable of interest, in combination with tumor size, but have had variable instances of residual disease at the time of definitive surgery, therefore questioning the appropriate selection of patients for preoperative excisional procedures. The novelty of our study lays in the aim to focus on those who had an excisional procedure and had no residual disease at the time of final hysterectomy. In doing so, we selected the ideal population to look at the safety of surgical approach, and we did not find any difference. Therefore, in cases where conization margins are negative for invasive disease, it may represent a group of patients where a minimally invasive approach to surgery may be reasonable without increased oncologic risk, and as such may lead to minimizing surgical risks associated with open procedures. However, due to the low event rates these findings must be interpreted with caution. Further prospective data are warranted to investigate the impact of preoperative conization in a carefully selected population to determine the safety of surgical approaches.

## 5. Conclusions

Our study demonstrated that in patients with a prior excisional procedure and no residual disease on hysterectomy specimen, the surgical modality did not appear to have an impact on RFS or OS. However, this conclusion is based on a small number of events in a retrospective study, and therefore future studies are warranted, taking into account which patients with negative margins may be treated with MIS surgery.

**Author Contributions:** C.A.: Data curation, conceptualization, drafting original manuscript, editing. G.R.P.: formal analysis, writing, review, editing. L.H., D.V.: 4C lead, conceptualization, analysis, writing review, editing. L.E., M.P., S.L., J.S.K., A.D.A., K.W., T.F., J.S., V.S.: conceptualization, writing review, editing. L.B., N.C., N.-B.S., L.-A.T.-F., S.P., S.R.K., N.S., J.-H.J., S.S.: data curation, writing review, editing. G.N.: conceptualization, drafting original manuscript, analysis editing. All authors have read and agreed to the published version of the manuscript.

**Funding:** This research received no external funding.

**Institutional Review Board Statement:** Institutional Review Board approval was obtained in each participating sites of 4C (McMaster University, University of Toronto, University of Calgary, University of Alberta, University of British Columbia, University of Manitoba, McGill University, University of Montreal, Laval University, and Dalhousie University). The study was conducted in accordance with the Declaration of Helsinki, and REB approval specific to the first and last authors of this study were obtained through approved by the Health Research Ethics Board of Alberta- Cancer Committee of University of Calgary (HREBA-CC 19.0350, 1 October 2019).

**Informed Consent Statement:** Patient consent was waived due to this being a retrospective study.

**Data Availability Statement:** Data is stored on an institutional secure REDCap database, complying with local REB. Raw data can be available on reasonable request.

**Conflicts of Interest:** G.N. has received speaker feeds from GSK, Pfizer, 3M, and Medtronic, and serves as the treasurer for the ERAS Society. A.D.A. has received research support from AstraZeneca, Pfizer, Cloivis, CCMB foundation, CCTG, and Merck, and has received speaker fees from AstraZeneca and Merck. Additionally is on the GSK and AstraZeneca advisory board, and on the board of directors for GOC, DocsMB, and University medical group. L.H. has received speaker fees from MSD, and serves on the Advisory board for MSD. She also serves as a board member of the Israeli Society of Gynecologic Oncology. J.S.K. has received research grants to her institution from CIHR, and CCS, and has also received the Michael Smith Foundation for health research Clinician scientist award. She has also received speaker feeds from AstraZeneca, and serves on the editorial board for Gynecologic Oncology, and is the Principle editor for the Journal of Gynecologic Oncology. G.R.P. has received consulting fees from Merck, and Profound medical, as well as honoraria from AstraZeneca. He is also involved in the Takeda Board, and declares a close family member to be an employee at Roche Ltd. None of the these conflicts of interest are felt to be directly relevant to this manuscript, and the remainder of the authors have any relevant conflicts of interest.

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
