# Peer review of "Oncologic Outcomes of Surgically Treated Cervical Cancer with No Residual Disease on Hysterectomy Specimen: A 4C (Canadian Cervical Cancer Collaborative) Working Group Study"

_curroncol, doi:10.3390/curroncol30020153_

Round 1
Reviewer 1 Report
Thank you for this interesting paper which is of interest to the Canadian Gyn Onc community.
I have no major concerns/comments.
I do note that the adenocarcinoma rate was high at 38% and there was a 9 % difference in AH vs LH within adenocarcinomas. Is the rate higher than expected? could they have been diagnosed with larger excisions than Squamous cancers? could this interfere with results.
Author Response
Yes, this rate is higher than expected for adenocarcinomas within this cohort. In addition you are correct to infer that the difference in surgical approach although not statistically significant within the analysis for histology, seems to be larger than you would expect. Certainly this could play into these results, but again, with the sample size it is limited. I have added this into our results section accordingly.
Reviewer 2 Report
In the manuscript titled 'Oncologic outcomes of surgically treated cervical cancer with no residual disease on hysterectomy specimen: a 4C (Canadian Cerbical Cancer Collaborative) working group study' the authors performed a large, multi-institutional collective effort to identify consecutive cases of surgically treated cervical carcinoma in Canada from December 2007-January 2017. The dataset included patient-level data and reflected a period of time when a variety of surgical approaches were used in the treatment of early-stage cervical disease. Cases were individually identified and raw data were extracted from the chart and pathology reports. Additionally, the data set was queried as a whole, and individual cases requiring further clarification were available. In focusing on cases who underwent conization, and then were followed by hysterectomy where no residual disease was identified, the authors further characterized this low-risk population.
Overall the study presented interesting data in alignment with relevant literature. The authors have described the limitations of the study in detail as well. Therefore, the manuscript can be recommended for publication in Current Oncology.
Minor comment:
There are language-related errors throughout the manuscript and must be proofread. In the title, the word 'cervical' is misspelled.
Author Response
Errors within the manuscript have been addressed.
Reviewer 3 Report
Dear Author’s
I was pleased to review your interesting article and i have the following comments:
Please do not repeat the different information belong the different se tion of the article “ introduction, methods, discussion (for ex: tge study period 2007-2017…).
PLease explain how the patients with invasive dissease were candidated for local excision prior hysterectomy?
Please explain the novelty of the study.
There are multiple studies in the literature about the ideal surgical approach “laparoscopy versus open surgery” for cervical cancer.
Knowing this results the minimal invasive surgery has the same outcomes in the treatment of cerv cancer.
Minimal english edits - please verify the title.
Author Response
Please do not repeat the different information belong the different se tion of the article “ introduction, methods, discussion (for ex: tge study period 2007-2017…)
This has been addressed and removed
PLease explain how the patients with invasive dissease were candidated for local excision prior hysterectomy?
We did not have the indication for LEEP procedure recorded. However, they did all have a diagnosis of cervical carcinoma prior to surgery. This has been added into the methods section.
In the limitiations section there is a statement that the LEEP indication was not recorded.
Please explain the novelty of the study.
See final paragraph of results section, re-worded to make this a stronger point
There are multiple studies in the literature about the ideal surgical approach “laparoscopy versus open surgery” for cervical cancer.
Knowing this results the minimal invasive surgery has the same outcomes in the treatment of cerv cancer.
yes- our study is novel in that we looked at a very low risk population, who had a LEEP procedure (presumably for appropriate indications, as there were no final residual cancer on the hysterectomy specimen, and we can therefore use this low risk population to look at the safety of surgical approach, and nodal risk, and work backwards to shed more light on who are appropriate candidates for MIS surgery.
Minimal english edits - please verify the title.
Done
Reviewer 4 Report
This is a well-organized paper that describes the oncologic outcomes in cases of cervical cancer initially treated with conization, and subsequently found to have no residual cervical cancer after hysterectomy performed via open and minimally invasive approaches.
I have only one question. In this study, 43 patients underwent a simple hysterectomy. What were the indications for simple hysterectomy?
Author Response
I have only one question. In this study, 43 patients underwent a simple hysterectomy. What were the indications for simple hysterectomy?
The indications are not known in this study as this information was not recorded. That said- 55 patients has stage 1A1 with LVSI, and there is an increasing movement for less radical surgery within the earlier stage cervical cancers, that is site-specific within the study population. There were many sites also that were also sites recruiting for the SHAPE trial within the time frame of this study.
Reviewer 5 Report
The impact of preoperative conization as a protective maneuver has been proposed. It is hypothesized that preoperative excisional procedures may translate into more favorable outcomes, both in the minimally invasive and open surgical modalities. The authors postulated that in cases where conization margins are negative for invasive disease, it may represent a group of patients where a minimally invasive approach to surgery, may be reasonable without increased oncologic risk. However, the present analysis has an important limitation for the low-risk group of patients, as reported by the percentage of lymph node metastases (less than 2%), the small tumor volume and consequently only two recurrences are reported.
Previous propensity-score matching study showed that conization before radical hysterectomy is associated with better oncological outcomes (PMID: 33620615).
In the end, what can we take from these data? A few things should be recognized, which include that surgery, even with minimally invasive approach in very select patients is feasible and safe, although in select centers and by select surgeons proficient in such an approach. moreover, the risk for over treatment considering the morbidity should be included in the therapeutical strategy. Also regarding the indication to perform a systematic lymphadenectomy (PMID: 28608117)
Author Response
In the end, what can we take from these data? A few things should be recognized, which include that surgery, even with minimally invasive approach in very select patients is feasible and safe, although in select centers and by select surgeons proficient in such an approach. moreover, the risk for over treatment considering the morbidity should be included in the therapeutical strategy.
This is also added into the final paragraph of the discussion.
Also regarding the indication to perform a systematic lymphadenectomy (PMID: 28608117)
This reference and description highlighting this was added to the discussion
Round 2
Reviewer 3 Report
Thank you for your revised manuscript.